# Microwave-Assisted Extraction Optimization and Effect of Drying Temperature on Catechins, Procyanidins and Theobromine in Cocoa Beans

**DOI:** 10.3390/molecules28093755

**Published:** 2023-04-27

**Authors:** Yessenia E. Maldonado, Jorge G. Figueroa

**Affiliations:** Departamento de Química, Universidad Técnica Particular de Loja (UTPL), Calle Marcelino Champagnat s/n, Loja 110107, Ecuador

**Keywords:** *Theobroma cacao* L., procyanidin B2, response surface methodology, microwave-assisted extraction, HPLC-DAD-ESI-IT-MS

## Abstract

Cocoa beans (*Theobroma cacao* L.) are an important source of polyphenols. Nevertheless, the content of these compounds is influenced by post-harvest processes. In this sense, the concentration of polyphenols can decrease by more than 50% during drying. In this study, the process of procyanidins extraction was optimized and the stability of catechins, procyanidins, and theobromine to different drying temperatures was evaluated. First, the effectiveness of methanol, ethanol, acetone, and water as extract solvents was determined. A Box–Behnken design and response surface methodology were used to optimize the Microwave-Assisted Extraction (MAE) process. The ratios of methanol-water, time, and temperature of extraction were selected as independent variables, whereas the concentration of procyanidins was used as a response variable. Concerning the drying, the samples were dried using five temperatures, and a sample freeze-dried was used as a control. The quantitative analyses were carried out by HPLC-DAD-ESI-IT-MS. The optimal MAE conditions were 67 °C, 56 min, and 73% methanol. Regarding the drying, the maximum contents of procyanidins were obtained at 40 °C. To our knowledge, this is the first time that the stability of dimers, trimers, and tetramers of procyanidins on drying temperature was evaluated. In conclusion, drying at 40 °C presented better results than the freeze-drying method.

## 1. Introduction

Ecuadorian cocoa (*Theobroma cacao* L.) is a product highly appreciated worldwide for its content of polyphenol compounds [1], whose concentration in beans ranges between 12% to 15% by dry weight [2]. The following polyphenols are found in cocoa beans: catechins (catechin and epicatechin), counting for about 37%; and anthocyanins and procyanidins, with 4% and 58%, respectively [3]. Some polyphenolic compounds, such as anthocyanins, are mainly found in pigment cells, which are responsible for giving color to the fruit [1]. The pigmentation depends on the concentration of these metabolites, producing a color ranging from white to intense violet [1]. Cocoa beans not only have polyphenol compounds, but also some alkaloids, specifically methylxanthines [4], such as theobromine and caffeine, whose content can reach 4% [5]. Theobromine is the main alkaloid in cocoa; it is found in cocoa trees, seeds, and pods. The concentration of alkaloids depends on the post-harvest treatment, specifically fermentation. These two groups of natural products contribute to taste, the polyphenols conferring astringency and the alkaloids a bitter taste [6]. Furthermore, polyphenols are related to many healthy properties of cocoa, since they present important biological activities such as antioxidant, anti-inflammatory, and anti-cancerogenic, among others [7]. Due to all these beneficial properties, the quality of cocoa is closely related to the concentration of polyphenols in the final product, which mainly depends on genetic type, post-harvest handling (fermentation and drying), transformation (roasting and conching), and the cultivation soil [8,9]. For all these reasons, it is very important to determine the conditions that ensure the highest polyphenol stability in post-harvesting processes. In fact, post-harvesting affects the concentration of polyphenols in cocoa beans, as demonstrated in a study on the national Ecuadorian variety by Albertini et al. [9], where the effects of fermentation and drying on totals polyphenol contents were evaluated, together with the antioxidant capacity. On these bases, the present research seeks the stability of polyphenols in cocoa beans, as indicators of the antioxidant properties in the drying processes. In order to reach this objective, it is mandatory to previously investigate and optimize the extraction process, develop an analytical method, and subsequently apply them to cocoa samples, submitted to different drying conditions. In this study, the problem was faced by Microwave-Assisted Extraction (MAE), statistical elaboration through ANOVA, and Response Surface Methodology (RSM), followed by HPLC-DAD-ESI-IT-MS analyses. The present investigation focused on the CCN-51 variety (Castro Naranjal Collection), characterized by greater resistance to fungal diseases and adaptability to different types of soils, making it the preferred variety among growers [10].

Regarding drying stage, the beans present an approximate humidity of 60%; this humidity should be reduced to 7% to avoid the proliferation of fungi and unpleasant odors, which would diminish the quality of cocoa [11]. On the one hand, it is recommended that the humidity is reduced to a minimum of 6% or 7% and a maximum of 8%; otherwise, the bean becomes fragile [12]. On the other hand, the speed of drying is a crucial factor, since eliminating the moisture too slowly promotes some oxidation reactions [13]. At this stage, the content of polyphenol compounds decreases due to the action of high temperatures [14,15]. If these compounds undergo oxidation, they originate quinones and protein condensation [4]. Many studies have evaluated the effect of drying on the concentration of polyphenolic compounds. Among them, we can mention the case of research by Santhanam et al. [3], where the authors concluded that the content of polyphenols in Malaysian cocoa beans is dependent on the drying temperature and time, whereas freeze-drying is the most conservative method for this class of compounds.

In a study conducted by Hii et al. [16] on the kinetics of cocoa drying, they compared sun-dried and oven-dried cocoa beans and concluded that the polyphenol content had significant differences. The study by Alean et al. [15] developed a mathematical model, which helped to predict how drying time influences polyphenols and also found that there is a 45% loss of polyphenolic compounds during drying. In other cases, the interest was focused on the antioxidant properties of polyphenols from cocoa, whose extraction conditions were optimized through a surface response method [13]. In addition to cocoa, other vegetal species have been submitted to similar investigations. In fact, in a published study, Rodríguez-Pérez et al. [17] described the optimization of microwave-assisted extraction polyphenols from *Moringa oleifera*. In this research, the authors use a response surface methodology (RSM) to obtain the optimum conditions of their extraction process. Furthermore, they evaluate the antioxidant and anti-scavenging activity of the extracts as a variable related to the polyphenol content and extraction yield. The present work represents the first study about the effect of drying temperature on the content of procyanidins, whose antioxidant activity is higher than those of their monomers [18]. Finally, comparing in the literature the Ecuadorian cocoa with the one from other important producer countries, we can observe that its total polyphenol content is higher than cocoa from Venezuela and most of the African Countries, except Madagascar, but it is lower than the Mexican product [19]. From the agronomic point of view, the Ecuadorian cocoa presents the highest yield in terms of kg/tree respect to the product from Mexico and Guatemala (1.325 kg vs. 1.189 kg and 1.035 kg, respectively). Furthermore, it presents the lowest incidence of infection by Black Pod (6.7% vs. 23.4% and 24.1%, respectively). On the other hand, the Ecuadorian cocoa is characterized by lighter dry beans, with an average weight of 1.30 g vs. 1.70 g and 1.50 g for the Mexican and Guatemalan cocoa, respectively [20].

## 2. Results and Discussion

### 2.1. Identification of Compounds in Cocoa Beans by HPLC-DAD-ESI-IT-MS

The High-Performance Liquid Chromatography (HPLC) chromatogram, recorded at a wavelength of 280 nm, for the cocoa bean microwave-assisted extraction (MAE) extracts, is presented in Appendix A. The peaks within the chromatogram have been assigned a numerical value corresponding to their order of elution. The identified compounds were characterized by comparing their retention time and MS and MS/MS spectra obtained through the mass analyzers to those of authentic standards, when available. Additionally, previously published literature was referenced to aid in the identification of these compounds.

The presence of theobromine, catechin, epicatechin, and procyanidin B1 and B2 in the cocoa bean MAE extracts were unequivocally confirmed, as their retention time (15.43, 20.41, 24.07, 17.67 and 21.09 min, respectively) and mass spectrometry (MS) data were found to be consistent with those of the corresponding commercial standards.

Furthermore, the procyanidin type B isomer 1 was also tentatively identified with *m*/*z* 577 and retention time (RT) 32.94 min. The main MS/MS fragments were *m*/*z* 425, 451, 407, and 289. The fragmentation pattern was consistent with previous report [21]. The ions *m*/*z* 425 and 407 correspond to a retro-Diels–Alder reaction (RDA) and a consecutive RDA reaction with a loss of a molecule of water. The ions at *m*/*z* 451 result from the elimination of the phloroglucinol molecule (A-ring) [21]. Moreover, the ion at *m*/*z* 289.072 corresponds to a loss of an (epi)-catechin unit.

In addition, two procyanidins type B trimers were also experimentally identified with *m*/*z* 865 and RT 24.91 and 32.00 min. The major MS/MS product ions were *m*/*z* 577, 739, 695, and 713. The ion at *m*/*z* 577 responds to a quinone methide (QM) cleavage. Furthermore, the observed fragment *m*/*z* 739 was a result of heterocyclic ring fission (HRF). Finally, the ion at *m*/*z* 695 is a relevant product resulting from the dehydration of the *m*/*z* 713 ion, which occurs subsequent to the RDA cleavage of the heterocyclic ring system.

Peaks with RT 21.59, 27.11, 29.20, and 36.18 min were tentatively identified as procyanidin type B tetramers. These peaks showed a precursor ion [M − H]^−^ 865 and the main MS/MS fragments were *m*/*z* 577, 739, 695, and 713. The ions at *m*/*z* 865 and 575 were a result of QM cleavages. Meanwhile, the fragment ion at *m*/*z* 983 was a result of water elimination after the RDA reaction. Moreover, the ion at *m*/*z* 1027 was derived from the heterocyclic ring fission. Finally, compound **12** with RT 36.18 min was also tentatively identified also a procyanidin type B tetramer. Nevertheless, the identification was made based on the positive ESI mode. This compound shows precursor ion [M − H]^+^ 1155.

### 2.2. Solvent Selection

The solubility of polyphenols depends on their molecular polarity and on the presence of functional groups, specifically hydroxyl and carbonyl groups, which can establish hydrogen bonds. In addition, hydroxyl groups can be conjugated with sugars or be alkylated, which means that a single extraction process for this class of metabolites does not exist [22]. For this reason, the selection of the solvent is decisive to obtain a good extraction yield. The most described organic solvents in the literature for the extraction of polyphenolic compounds are acetone, methanol, ethanol, and water [23]. A series of extraction experiments with different solvents was carried out to determine the system that guarantees the highest yield in polyphenols and theobromine extraction.

Figure 1 shows that the solvent mixture methanol:water (1:1) gave the best extraction yield for most of the compounds, such as in studies about polyphenols by Machado et al. [24] on Brazilian guarana; Mokrani and Madani [25] on peach; and Lončarić et al. [26] about blueberry pomace. On the one hand, if taken separately, some analytes are best extracted with pure methanol. This is the case of procyanidins B2 and the tetramer B isomer 2. On the other hand, 100% methanol is equivalent to methanol/water for theobromine and epicatechin, which is consistent with the results obtained by Kallithraka et al. [27] on grape seeds. Similarly, a study by Naima et al. [28] indicates that the best solvent to extract polyphenols from *Acacia mollissima* barks by microwave digestion was methanol. Roughly 100% acetone is a relatively effective solvent for catechin, whereas pure ethanol is a rather unsatisfactory solvent for all the analytes of the present study. A study by Borges et al. [29] on *Euterpe edulis* pulp indicates that ethanol is the worst solvent for the extraction of total flavanols, which agrees with the present investigation. In contrast, the addition of water to ethanol drastically changes its properties, making it the best choice for extracting theobromine. This result is consistent with the study conducted by Machado et al. [24] on Brazilian guarana. Finally, 100% water is a moderately effective solvent for extracting both catechin and theobromine. Indeed, in a study by Wissam et al. [30] on the effective extraction of polyphenols and proanthocyanins from pomegranate peel, it is emphasized that water is an interesting solvent for the food and pharmaceutical industry because it is non-toxic and the yield it possesses is acceptable.

### 2.3. Optimization of the MAE Method

After determining that the mixture methanol:water (1:1) is the best solvent for the extraction of the compounds of interest, a response surface methodology (RSM) study was performed to optimize the extraction conditions of polyphenols. In this sense, the methanol concentration in the methanol:water mixture, temperature, and extraction time were selected as independent variables.

The sample presented a high concentration of procyanidins (dimers, trimers, and tetramers), which are very thermolabile compounds [31]. Moreover, considering that procyanidins are formed by the binding of one or more molecules of catechin and/or epicatechin [32], the increase in catechin concentration can be attributed to the degradation of these compounds. It is for this reason that catechin and epicatechin were not considered for the optimization of the experimental conditions and they are not represented in Figure 2.

In Appendix A, the Pareto diagrams of the standardized effects for each response variable are presented. For type A procyanidin dimers, only the concentration of methanol in the extraction solvent was found to have an effect, demonstrating increased extraction with higher concentrations of methanol. Regarding type B procyanidin dimers, both the temperature and extraction solvent showed linear and quadratic effects, as evidenced in Figure 2. The best results for this type of procyanidin were observed with temperatures between 60–80 °C and methanol concentrations close to 80%. Additionally, for trimers and tetramers procyanidins, a quadratic effect was observed for the extraction solvent and a linear effect for extraction temperature. For these two types of procyanidins, the best extraction was achieved with a methanol concentration close to 70% and an extraction temperature of 65 °C.

The impact of temperature and time on the content of polyphenol compounds is shown in Figure 2a, where it can be observed that the extraction time has no relevant effect on the yield of the analytes coinciding with the studies carried out on nettle leaves by Vajić et al. [33]. Otherwise, temperature plays a fundamental role when it is in the range of 60 to 70 °C, coinciding with the studies carried out on garlic by Ciric et al. [34], and on grape stalks and marc by Spigno and De Faveri [35]. The authors indicate that the extraction of polyphenol compounds at 60 °C is more effective than at lower temperatures. Nevertheless, temperatures higher than 70 °C produced a degradation of dimers B, trimers B, and tetramers B. Furthermore, the dimer A-type presented a higher resistance to temperature than the B-type; this could be due to double linkage consisting of a C-C single bond and an additional ether bond [31]. 

In Figure 2b, the relationship between temperature and methanol concentration is observed, indicating the increase in extraction yield at temperature values below 100 °C and methanol concentrations around 60%. This trend coincides with a study conducted on the extraction of phenolic compounds in apples, where it is shown that, as long as water increases, the solvent is much more efficient [36]. In the case of type B dimers, the trend is quadratic, with the extraction yield increasing at 70 °C and 73% MeOH and then decreasing for higher values of temperature and solvent concentration. Dailey and Vuong [37] observed similar behavior with temperature in the extraction of phenolic compounds from macadamia.

Figure 2c shows that in the relationship between time and methanol concentration, an increase in the extraction yield is correlated to the increase in solvent concentration. Indeed, a maximum increase in the extraction yield of the analytes is observed, corresponding to the composition of methanol at approximately 73%. These data are consistent with what is reported in the literature, where it turns out that aqueous solvents are more effective than pure solvents [36]. Time does not have a significant effect on the extraction yield, although an optimal value of 56 min was determined, consistent with a study by Wiyono et al. [38].

The model mathematics for the dependent variables is presented in Appendix A. The models are capable of explaining more than 77% of the variance in the data. The optimal conditions obtained for the extraction of polyphenols from cocoa beans are 73% MeOH and 67 °C for 56 min. These results are supported by a study conducted by Dewi et al. [39] on cocoa pod husk.

Other compounds that impact the sensory profile of cocoa beans have been identified, such as citric acid and theobromine [40]. To provide complementary information, optimal conditions were determined for extracting these compounds. The results showed that the optimal extraction of both compounds was achieved using an 8.1% methanol solution at 114 °C for 60 min. The concentration of methanol required to extract citric acid and theobromine decreased considerably compared to procyanidins, due to the greater solubility of these compounds in water, especially theobromine. Regarding the extraction time, a negative quadratic effect was observed only for citric acid, whereas no effect was observed for theobromine. In addition, both compounds exhibited lower thermolability. Due to this thermal stability, it was not possible to define unique conditions that would allow the maximization of extraction of all compounds. In fact, if procyanidins, citric acid, and theobromine are defined as response variables, a low overall desirability of 0.6 is obtained.

On the other hand, the optimal conditions for extracting theobromine were found to be 45.45% MeOH at 150 °C for 20 min. However, these compounds were not included as response variables in the study because of their high thermal stability, which resulted in a low overall desirability of 0.6.

### 2.4. Effect of Drying on the Polyphenol Content in Cocoa Beans

After the fermentation stage, cocoa has a moisture content of about 60%, which makes the plant matrix susceptible to decomposition [41]. In this sense, to increase the stability of the product and prevent over-fermentation, it is essential to decrease the water content progressively, until reaching an optimal moisture content of 8% [12]. The kinetics of the drying process for CCN-51 cocoa beans, at different temperatures, are shown in Figure 3.

According to Figure 3, a proportional relationship between temperature and drying rate can be observed. For example, at a temperature of 70 °C, the time required to reach half of the moisture content is about 1/3 concerning the temperature of 30 °C. Two stages in drying were also observed. In the first stage, the moisture ratio lowers rapidly, reaching approximate values of 0.25, 0.20, 0.17, 0.10, and 0.05 at 30, 40, 50, 60, and 70 °C, respectively. In the second stage, the trend is almost linear until the constant weight of the beans. It can be observed that the constant weight is reached on approximately 23, 27, 38, 40, and 41 h at 70, 60, 50, 40, and 30 °C, respectively. The trend of the drying curves represented in Figure 3 is consistent with what has been described in the literature with other plant products [42]. A study by Faborode et al. [43] on the degradation of polyphenolic compounds in cocoa shows that at temperatures above 60 °C, cocoa quality is not optimal. The same information is reported in a study by Kyi et al. [44] on Malaysian cocoa beans, indicating that the higher the drying temperature (more than 60 °C), the greater the degradation of polyphenols. Furthermore, a slow drying process also promotes the degradation of polyphenolic compounds by cell destruction, as described in the literature [14,15].

The content of polyphenols and theobromine in cocoa beans after drying is represented in Figure 4. For most of the procyanidins, there was a decrease in the concentration between 15 to 25% as the drying temperature increased. Furthermore, it was observed that an elevation in temperature from 40 to 70 °C resulted in a more pronounced degradation of compounds containing a larger number of linked monomers. Specifically, the mean degradation for dimers, trimers, and tetramers were 12.6%, 19.5%, and 21.5%, respectively.

On the contrary, the catechin concentration increased more than two times as the drying temperature increased. The increase observed for catechin at a high temperature can be explained by the formation of this polyphenol as a dimer degradation product [31].

The quantitative results by HPLC indicated that the polyphenol stability is maximized at 40 °C during the drying process. Furthermore, in many cases with this temperature, the concentration of epicatechin and procyanidins was higher than that found in the freeze-dried sample. This can be attributed to the drying time, being 40 and 68 h for the sample dried with hot air at 40 °C and freeze-dried, respectively. The reduction of polyphenols in the sample freeze-dried could be due to the result of oxidative reactions [8,44]. Moreover, the hot air (40 °C) drying method is more economical than freeze-drying.

Regarding theobromine, it is the second most abundant analyte in cocoa beans, as mentioned by Carrillo et al. [45]. Drying did not cause a major change in its concentration, presenting higher values when the cocoa beans were dried at 60 °C. With these conditions, there was no difference in the amount of analyte found in the control sample (freeze-dried).

Alean et al. [15] determined both experimentally and through a mathematical model, that the optimum temperature minimizing the total polyphenols decomposition in beans of CCN-51 cocoa is 40 °C, although they worked with the Folin–Ciocalteu method instead of HPLC. More specifically, in a study by Payne et al. [46], the concentration of epicatechin and catechin in cocoa beans, after drying at 45 °C, was determined to be about 9500 and 300 mg/kg, respectively. These values are very close to the results obtained in the present investigation. Regarding procyanidins individually, no data have been found on cocoa beans. However, in an investigation conducted by Khanal et al. [47] about blueberry procyanidins, the tendency here described was confirmed. Finally, in regard to theobromine, a study by Hernández-Hernández et al. [48] on cocoa beans dried at 55 °C reported that this alkaloid reached a concentration of about 6600 mg/kg. This value is the closest to the one obtained in the present study among the data from the literature. The difference can be attributed to geographic, genetic, and climatic factors [3].

## 3. Materials and Methods

### 3.1. Solvents and Standards

The hexane used in defatting was a reagent-grade solvent, purchased from Fisher Chemical (Madrid, Spain). The solvents of the mobile phase (acetonitrile and water) were HPLC grade, purchased from Panreac Applichem (Barcelona, Spain). All the analytical standards (theobromine, catechin, epicatechin, procyanidin B1, procyanidin B2, and procyanidin A2) were supplied by Sigma-Aldrich (St. Louis, MO, USA) or LGC Standards (Wesel, Germany). Acetone and ethanol for extraction were HPLC grade and purchased from Merck KGaA (Darmstadt, Germany), whereas methanol was supplied by Panreac Applichem (Barcelona, Spain).

### 3.2. Vegetal Material

The samples of cocoa *T. cacao* L were collected in the Ventanas canton, province Los Ríos, Ecuador, from the cultivated CCN-51 variety. The samples were washed with abundant water to remove any impurities. They were then left to stand for 4 days so that the mucilage would detach from the cocoa bark, which indicates a correct maturity according to Enríquez [49]. Finally, the pods were cut transversally without damaging the beans, which were removed and anaerobically fermented at room temperature. The fermentation was carried on for 48 h, in 1 L flasks, with several beans ranging between 878 and 1040 g approximately.

### 3.3. Sample Preparation 

A control sample was prepared by freeze-drying the beans that were dehydrated directly and applying them to a freeze-dryer apparatus (FreeZone 12, Labconco Corporation, Kansas City, MO, USA). After freeze-drying, the beans were milled in an ultracentrifuge knife-miller (ZM200, Retsch, Dusseldorf, Germany) until obtaining a particle size of 500 μm. Since in cocoa beans, the total fat content exceeds 50% on a dry basis [50], the fat fraction interferes in the extraction process, causing a low recovery of polyphenolic compounds. For this reason, a method was applied for defatting the sample before extraction. The defatting process was carried out by weighing 2 g of sample in centrifuge tubes and mixing them with 20 mL of hexane. The mixture was shaken for 30 min at 200 rpm in a horizontal shaker and subsequently centrifuged for 20 min at 8000 rpm in a centrifuge (Sorvall ST8, Thermo Scientific, Waltham, MA, USA). The sample was then recovered, and the supernatant was discarded. Finally, the samples were placed under a vacuum for 15 h to eliminate the residual hexane.

### 3.4. Solvent Selection and Extraction Optimization

The extraction solvent was selected, submitting the defatted samples to microwave-assisted extraction (MAE) in a laboratory oven (Mars 6, CEM Corporation, Matthews, NC, USA). The solvents used were acetone (Me_2_CO), water, ethanol (EtOH), methanol (MeOH), acetone/water 50%, ethanol/water 50%, and methanol/water 50%. The extraction was carried out by suspending the defatted samples in 40 mL of solvent in Teflon tubes. The oven was programmed to maintain a constant temperature of 100 °C for a period of 30 min, during which the power output was set to 1500 W.

Once the best solvent system was determined, the MAE conditions were optimized. The optimization was carried out using a Box–Behnken design, as detailed in Table 1. In all experiments, the microwave power was configured to operate at 1500 W. The concentrations of procyanidins B2, isomer 1 of trimer B, and isomer 2 of tetramer B were used as response variables. These compounds were selected based on their predominance in their respective classes. In contrast, the concentrations of catechin and epicatechin were not selected as response variables due to their role as the monomeric units of procyanidins. Therefore, any increase in their concentrations could be attributed to the degradation of procyanidins rather than to higher extraction. Furthermore, the concentrations of citric acid and theobromine were excluded from the selection process due to their relatively lower thermolability compared to procyanidins, as well as their opposite water solubility properties. In particular, theobromine exhibits low solubility in water at room temperature.

The obtained extracts were immediately cooled in ice to attain a temperature of 20–25 °C and in the same conditions described in Section 3.3 and stored at −20 °C until use.

### 3.5. Effect of Drying Temperature on the Concentration of Procyanidins

The fermented beans were dried at 30 °C, 40 °C, 50 °C, 60 °C, and 70 °C until reaching a water content <10% by weight in a hot air dryer (DY-330H, Daeyeong E&B, Ansan-city, Republic of Korea). The required time was measured for each temperature. After that, the dried beans were submitted to milling, defatting, optimal MAE, centrifugation, and filtration as previously described. The filtered solutions were then stored at 4 °C until use.

### 3.6. Identification and Quantification of Polyphenols and Theobromine

Prior to analysis, the extracts were filtered with regenerated cellulose syringe-filters of 0.2 μm pore size (Millipore, Bedford, MA, USA).

Analyses were performed using a HPLC system (Dionex UltiMate 3000, Thermo Fisher Scientific, Waltham, MA, USA) coupled to an ion tramp mass spectrometer (AmaZon speed, Bruker, Billerica, MA, USA), equipped with an ESI (Electron Spray Ionization) ion source (Bruker, Billerica, MA, USA). The separations were performed through a C18 reversed-phase column (Acclaim TM 120, Thermo Fisher Scientific, Waltham, MA, USA), 250 mm long, 4.6 mm internal diameter, and with a fixed phase of 5 μm particle size. The elution was performed with a constant flow rate of 0.5 mL/min, using an acetonitrile/water (MeCN/H_2_O) mixture as a mobile phase, according to a decreasing polarity gradient.

The process started with 5% MeCN for 3 min, followed by an increase in the percentage of MeCN to 18% at 10 min. This composition was maintained until minute 19; from there, it went to 21% MeCN in 1 min. Subsequently, the MeCN concentration increased to 30% at minute 50 and to 60% at minute 60. Finally, after 1 min at constant composition, the percentage of MeCN decreased to 5% at minute 66.

The chromatograms with the photodiode array detector were obtained by measuring the absorbance at 280 nm.

The quantification was performed using a calibration curve for each of the following standards: theobromine, procyanidin B1, catechin, procyanidin B2, epicatechin, and procyanidin A2. Each curve was based on 10 points and presented an R^2^ ≥ 0.995. 

### 3.7. Statistical Analysis

The statistical analyses, one-way ANOVA, Tukey’s multiple range test, and RSM method were carried out using the software Minitab 16 (Minitab LLC, State College, PA, USA). On the one hand, ANOVA and Tukey’s multiple range test, with a significance of 0.05, were used to determine the effect of the drying conditions on the procyanidin concentration. On the other hand, the RSM was used to optimize the extraction method.

## 4. Conclusions

In the present study, the stability of procyanidins in the beans of the CCN-51 cocoa variety was analyzed, for the first time, under different drying conditions. Firstly, the best extraction solvent for major polyphenols and theobromine was determined to be aqueous methanol. After that, the optimum microwave-assisted extraction conditions were investigated, with 73% MeOH, 67 °C, and 56 min being the optimum parameters. Finally, the optimum dehydration temperature was established at 40 °C.

## Figures and Tables

**Figure 1 molecules-28-03755-f001:**
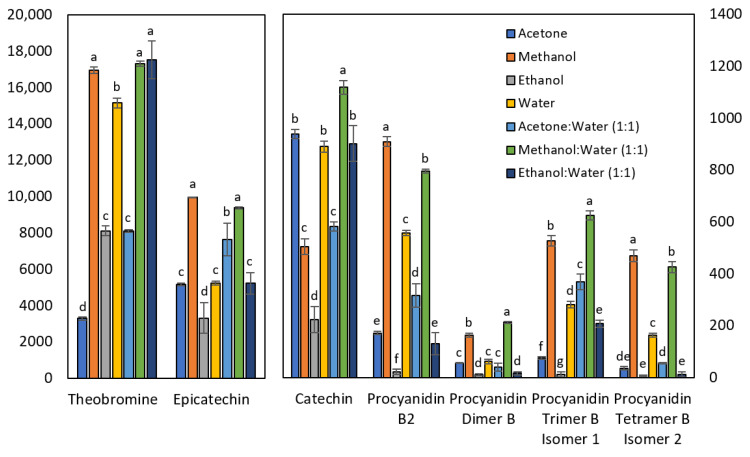
Effect of extraction solvent on the concentration of theobromine and polyphenolic compounds. Results are expressed in mg/kg. The mean values of each compound that do not share the same letter are significantly different.

**Figure 2 molecules-28-03755-f002:**
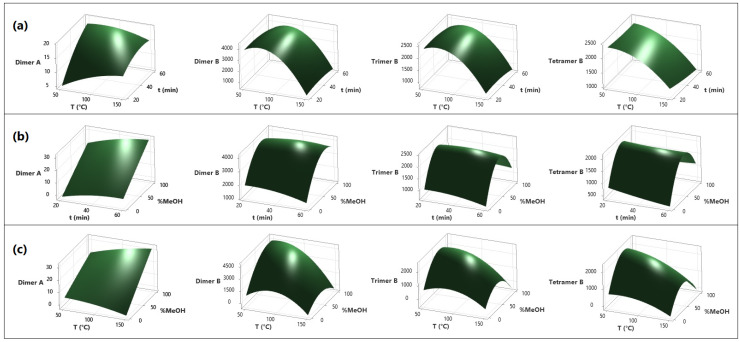
RSM of the microwave-assisted extraction of polyphenols with aqueous methanol. Variable interaction: (**a**) temperature vs. time, (**b**) time vs. MeOH concentration, and (**c**) temperature vs. MeOH concentration. The variables are time (min), temperature (°C), MeOH concentration (%), and analyte concentration (mg/kg).

**Figure 3 molecules-28-03755-f003:**
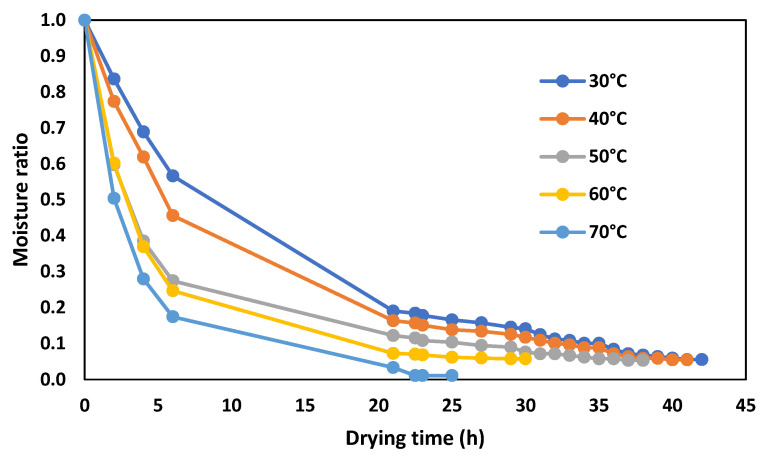
Kinetics of the drying process for CCN-51 cocoa beans at different temperatures.

**Figure 4 molecules-28-03755-f004:**
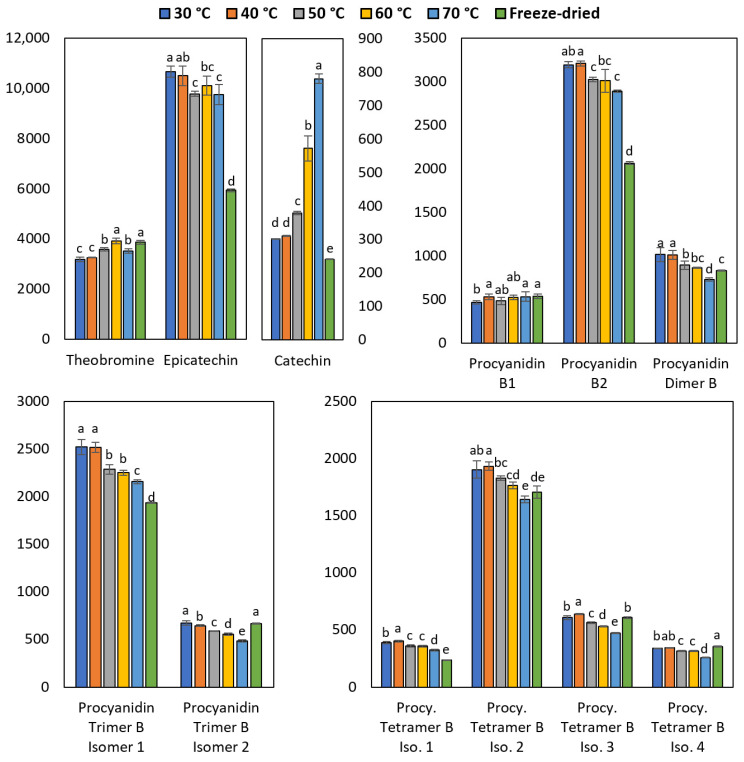
Effect of drying temperature on the concentration of theobromine and polyphenols. Results are expressed in mg/kg. The small letters over the bars indicate significant differences between the means according to ANOVA and Tukey tests (*p* < 0.05). The letter “a” represents the highest value. Error bars show the standard deviation of the mean.

**Table 1 molecules-28-03755-t001:** Box–Behnken design with natural and coded values for procyanidins extraction conditions.

Run	Temperature (°C)	Time (min)	MeOH (%)
1	100 (0)	40 (0)	50 (0)
2	100 (0)	20 (−1)	0 (−1)
3	150 (1)	60 (1)	50 (0)
4	50 (−1)	60 (1)	50 (0)
5	150 (1)	40 (0)	0 (−1)
6	50 (−1)	40 (0)	0 (−1)
7	50 (−1)	40 (0)	100 (1)
8	100 (0)	40 (0)	50 (0)
9	100 (0)	60 (1)	0 (−1)
10	150 (1)	20 (−1)	50 (0)
11	50 (−1)	20 (−1)	50 (0)
12	100 (0)	40 (0)	50 (0)
13	150 (1)	40 (0)	100 (1)
14	100 (0)	60 (1)	100 (1)
15	100 (0)	20 (−1)	100 (1)

## Data Availability

Data are available from the authors upon reasonable request.

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
