# Peer review of "Microwave-Assisted Extraction Optimization and Effect of Drying Temperature on Catechins, Procyanidins and Theobromine in Cocoa Beans"

_molecules, 2023, doi:10.3390/molecules28093755_

Round 1

Reviewer 1 Report

This manuscript is about the evaluation of drying parameters in the procyanidins of cocoa beans, but I have some considerations:

- The numeration of materials and methods is 4, but actually is 3

- The whole idea of the work is good, but it seems to me a little confusing. it seems that the whole idea was about the extraction of polyphenols was more important, but in fact was an analytical study to investigate the drying process, and how the polyphenols could be stable in different drying temperatures, is that right? I suggest rethink the introduction to explore better what is the difficulty of analyse these changes in polyphenols, because the work is much more analytics than cocoa drying

- I missed more information about MAE, what power did you use?

- Please improve your conclusion, it is not enriching your work

Author Response

Response to Reviewer # 1

Comments and Suggestions for Authors

Comment: “The numeration of materials and methods is 4, but actually is 3”

Response: The authors thank the reviewer for this observation. The identified mistake has been corrected.

Comment: “The whole idea of the work is good, but it seems to me a little confusing. it seems that the whole idea was about the extraction of polyphenols was more important, but in fact was an analytical study to investigate the drying process, and how the polyphenols could be stable in different drying temperatures, is that right? I suggest rethink the introduction to explore better what is the difficulty of analyse these changes in polyphenols, because the work is much more analytics than cocoa drying”

Response: The authors express their gratitude for your observation. You are correct, and as a result, we have modified the abstract and introduction to address this issue and prevent any further confusion.

Comment: “I missed more information about MAE, what power did you use?”

Response: We thank Reviewer #1 for his/her comment. In all experiments, the microwave power was configured to operate at 1500 W. This information was included in section 3.4

Comment: “Please improve your conclusion, it is not enriching your work”

Response: The authors have considered this suggestion and subsequently modified the title.

Reviewer 2 Report

Title

I suggest the authors include theobromine and catechins in the title because these analytes are considered a significant part of the manuscript. 

Abstract

Include in the abstract Theobroma cacao L., after Cocoa beans, and then the author can be abbreviated like T. cacao, the scientific name of the specie is misspelled. The authors wrote “Theobroma cocoa” this name doesn't exist. Please be careful with these mistakes.

The authors need to write Microwave Assisted Extraction (MAE), and then they can use MAE.

Why do the authors only use procyanidins concentration as the response variable? If the authors only use procyanidins concentration, they supposed that the catechins and theobromine behave similarly. Please, the authors need to clarify this situation.

When I read the authors' abstract, I needed clarification on the research question. I read a technical summary of an analytical chemistry practice. The importance of optimizing this methodology by MAE needs to be clarified why the metabolites they are monitoring are necessary and the pharmaceutical, food, or economic importance. The authors must rewrite their abstract where they make these aspects clear, and the reader finds it attractive to read the entire manuscript, not just based on saying that there is nothing described in the literature.

Introduction

What is the difference between Ecuadorian cocoa with Guatemala, Mexican, or African cocoa? It is essential to highlight this information. 

“Polyphenolic compounds are mainly found in pigment cells, which are responsible for giving color to the fruit” this sentence is ambiguous. The authors need to specify what type of polyphenolic compounds, anthocyanins, or which? 

“These two groups of natural products contribute to flavour and aroma, the polyphenols conferring the astringent flavour and the alkaloids conferring the bitter taste” Please, the correct form is "flavor," not "flavour". Are polyphenolic compounds contribute to aroma? Is it true? I think the aroma is related more to terpenes and essential oils. Please, clarify this idea and cite literature to support it.

“Furthermore, the one-by-one HPLC quantification of polyphenols is more accurate than the use of the FolinCiocalteu method” this sentence is evident. Chromatographic methods have more accuracy and selectivity, but what is the goal of highlighting this?

Figure 1 needs to be clarified. It is not defined what the intervals are, whether it is the standard deviation or a confidence interval. In addition, the letters on the bars need to be specified as to what they refer to.

In the section “solvent selection,” What were the MAE conditions? Were these conditions the central point (0, 0, 0) in Box–Behnken design? 

The section “Optimization of the MAE Method” is incomplete. The authors need to provide a Pareto Chart to discuss the effect of factors, if these are significant or not and if an interaction exists between these. Furthermore, it is mandatory when you develop an RSM to provide the general equation of the model and its derivatives equations.

Figure 4 has the same issues as figure 1. 

Why do the authors not use the central point in Box–Behnken design? (Tabla 1)

The authors mentioned Procyanidins Quantification through HPLC-DAD-IT-MS but only said that it quantified at 280 nm in UV. Please provide the ions or transitions in MS (SRM, MRM, or MSn) to quantify o; if the authors use the MS only as a reference to obtain the m/z of each compound, please provide this information in a table. The authors need to explain more about the quantification by UV and MS in results and provide chromatograms and MS spectrums. 

The conclusion only reflects the highlight of the results doesn't answer the research question or a proposal aim of their work. The authors need to rewrite.

The manuscript is fascinating. However, the authors provide an incomplete version of their work, and the aims are not clear. The research question is confusing. The authors need to improve their discussion of the manuscript and provide better conclusions.  

Author Response

Response to Reviewer # 2

Comment: Title: I suggest the authors include theobromine and catechins in the title because these analytes are considered a significant part of the manuscript.

Response: The authors have considered this suggestion and subsequently modified the title.

Comment: Abstract - Include in the abstract Theobroma cacao L., after Cocoa beans, and then the author can be abbreviated like T. cacao, the scientific name of the specie is misspelled. The authors wrote “Theobroma cocoa” this name doesn't exist. Please be careful with these mistakes.

Response: The authors thank the reviewer for this observation. We have carefully considered the reviewer's comments, resulting in the identification and subsequent correction of an error.

Comment: The authors need to write Microwave Assisted Extraction (MAE), and then they can use MAE.

Response: We appreciate the reviewer's feedback. The name of the technique has been added to the abstract.

Comment: Why do the authors only use procyanidins concentration as the response variable? If the authors only use procyanidins concentration, they supposed that the catechins and theobromine behave similarly. Please, the authors need to clarify this situation.

Response: The authors wish to extend their gratitude to the reviewer for highlighting this important point. As a result, an explanation has been included in Section 3.4.

“The concentrations of procyanidins B2, isomer 1 of trimer B, and isomer 2 of tetramer B were used as response variables. These compounds were selected based on their predominance in their respective classes. In contrast, the concentrations of catechin and epicatechin were not selected as response variables due to their role as the monomeric units of procyanidins. Therefore, any increase in their concentrations could be attributed to the degradation of procyanidins rather than to higher extraction. Furthermore, the concentrations of citric acid and theobromine were excluded from the selection process due to their relatively lower thermolability compared to procyanidins and their opposite water-solubility properties. In particular, theobromine exhibits low solubility in water at room temperature.”

Comment: When I read the authors' abstract, I needed clarification on the research question. I read a technical summary of an analytical chemistry practice. The importance of optimizing this methodology by MAE needs to be clarified why the metabolites they are monitoring are necessary and the pharmaceutical, food, or economic importance. The authors must rewrite their abstract where they make these aspects clear, and the reader finds it attractive to read the entire manuscript, not just based on saying that there is nothing described in the literature.

Response: The authors acknowledge the reviewer for their valuable contribution through this observation and offer their thanks. In light of this, the abstract has been revised and the research question clarified.

Comment: Introduction - What is the difference between Ecuadorian cocoa with Guatemala, Mexican, or African cocoa? It is essential to highlight this information. 

Response: We express our gratitude to Reviewer #2 for his/her valuable comment. The requested information has been added in section 1 - Introduction.

Comment: “Polyphenolic compounds are mainly found in pigment cells, which are responsible for giving color to the fruit” this sentence is ambiguous. The authors need to specify what type of polyphenolic compounds, anthocyanins, or which? 

Response: We appreciate Reviewer #2 for their comment. This information was clarified in the introduction.

Comment: “These two groups of natural products contribute to flavour and aroma, the polyphenols conferring the astringent flavour and the alkaloids conferring the bitter taste” Please, the correct form is "flavor," not "flavour". Are polyphenolic compounds contribute to aroma? Is it true? I think the aroma is related more to terpenes and essential oils. Please, clarify this idea and cite literature to support it.

Response: We appreciate Reviewer #3 for their observation, which we agree with. Indeed, these two groups of natural products contribute to taste, the polyphenols conferring astringency and the alkaloids a bitter taste. This aspect has been corrected in the manuscript in the introduction section.

Comment: “Furthermore, the one-by-one HPLC quantification of polyphenols is more accurate than the use of the FolinCiocalteu method” this sentence is evident. Chromatographic methods have more accuracy and selectivity, but what is the goal of highlighting this?

Response: We would like to express our sincere appreciation to reviewer #2 for their valuable observation. After careful consideration, we concur with their feedback, and consequently, we have taken prompt action to remove the sentence in question.

Comment: Figure 1 needs to be clarified. It is not defined what the intervals are, whether it is the standard deviation or a confidence interval. In addition, the letters on the bars need to be specified as to what they refer to.

Response: We appreciate Reviewer #2 for their comment. All the requested information has been added to Figure 1.

Comment: In the section “solvent selection,” What were the MAE conditions? Were these conditions the central point (0, 0, 0) in Box–Behnken design? 

Response:  We appreciate Reviewer #2 for their comment. Your observation is accurate, the central conditions of response surface design were utilized for the purpose of selecting the solvent.

The oven was programmed to maintain a constant temperature of 100°C for a period of 40 minutes, during which the power output was set to 1500 W.

Comment: The section “Optimization of the MAE Method” is incomplete. The authors need to provide a Pareto Chart to discuss the effect of factors, if these are significant or not and if an interaction exists between these. Furthermore, it is mandatory when you develop an RSM to provide the general equation of the model and its derivatives equations.

Response: We would like to express our gratitude to Reviewer #2 for their valuable feedback. As such, Pareto diagrams were included as a supplementary figure, and the analysis of each effect was incorporated into Section 2.2. Additionally, equations for each independent variable were provided in the supplementary materials.

Comment: Figure 4 has the same issues as figure 1. 

Response: We appreciate Reviewer #2 for their comment. All the requested information has been added to Figure 4.

Comment: Why do the authors not use the central point in Box–Behnken design? (Tabla 1)

Response: Dear reviewer, we appreciate your inquiry. Although if we use the central point, to clarify this aspect, Table 1 has been improved, including the Box-Behnken design with natural and coded values for procyanidins extraction conditions. The central points were runs 1, 8 and 12.

Comment: The authors mentioned Procyanidins Quantification through HPLC-DAD-IT-MS but only said that it quantified at 280 nm in UV. Please provide the ions or transitions in MS (SRM, MRM, or MSn) to quantify o; if the authors use the MS only as a reference to obtain the m/z of each compound, please provide this information in a table. The authors need to explain more about the quantification by UV and MS in results and provide chromatograms and MS spectrums.

Response: We appreciate Reviewer #2 for their observation. In this sense, the next information was detailed in the section 3.6.

The identified compounds were characterized by comparing their retention time, MS and MS/MS spectra obtained through the mass analyzers to those of authentic standards, when available. Additionally, previously published literature was referenced to aid in the identification of these compounds. Whereas, the quantification was performed by meas-uring the absorbance at a wavelength of 280 nanometers using a diode array detector.

Comment: The conclusion only reflects the highlight of the results doesn't answer the research question or a proposal aim of their work. The authors need to rewrite.

Response: We appreciate the reviewer for their constructive feedback on our conclusion section. We have carefully reviewed our writing and made significant changes to ensure that our conclusion clearly addresses the research question and aim of our study.

Comment: The manuscript is fascinating. However, the authors provide an incomplete version of their work, and the aims are not clear. The research question is confusing. The authors need to improve their discussion of the manuscript and provide better conclusions.  

Response: We would like to express our gratitude to the reviewer for their insightful comments, which have significantly contributed to the enhancement of this manuscript. The suggestions provided have helped us to clarify the objective of our investigation and substantially improve the discussion and conclusions. We hope that these revisions address the reviewer's concerns and improve the quality of our work. We thank the reviewer for their time and effort in reviewing our work.

Round 2

Reviewer 1 Report

No more comments

Reviewer 2 Report

The authors attended to all my comments, provided enough evidence in their answers, and improved the quality of their manuscript in the results and discussion section. Furthermore, the authors improved the introduction, highlighting their research's goal and research question. For that, I recommend publishing the manuscript with minor revisions, and the authors need to adapt their manuscript according to the journal's template.